# Phytofabrication of Silver Nanoparticles Using *Trigonella foenum-graceum* L. Leaf and Evaluation of Its Antimicrobial and Antioxidant Activities

**DOI:** 10.3390/ijms24043480

**Published:** 2023-02-09

**Authors:** Monika Moond, Sushila Singh, Seema Sangwan, Savita Rani, Anuradha Beniwal, Jyoti Rani, Anita Kumari, Indu Rani, Parvesh Devi

**Affiliations:** 1Department of Chemistry, CCS Haryana Agricultural University, Hisar 125004, India; 2Department of Microbiology, CCS Haryana Agricultural University, Hisar 125004, India; 3Department of Horticulture, CCS Haryana Agricultural University, Hisar 125004, India; 4Department of Plant Physiology, CCS Haryana Agricultural University, Hisar 125004, India

**Keywords:** silver nanoparticles, phytochemicals, antimicrobial, antioxidant

## Abstract

Silver nanoparticles (AgNPs) were fabricated using *Trigonella foenum-graceum* L. leaf extract, belonging to the variety HM 425, as leaf extracts are a rich source of phytochemicals such as polyphenols, flavonoids, and sugars, which function as reducing, stabilizing, and capping agents in the reduction of silver ions to AgNPs. These phytochemicals were quantitatively determined in leaf extracts, and then, their ability to mediate AgNP biosynthesis was assessed. The optical, structural, and morphological properties of as-synthesized AgNPs were characterized using UV-visible spectroscopy, a particle size analyzer (PSA), FESEM (field emission scanning electron microscopy), HRTEM (high-resolution transmission electron microscopy), and FTIR (Fourier transform infrared spectroscopy). HRTEM analysis demonstrated the formation of spherically shaped AgNPs with a diameter of 4–22 nm. By using the well diffusion method, the antimicrobial potency of AgNPs and leaf extract was evaluated against microbial strains of *Staphylococcus aureus, Xanthomonas* spp., *Macrophomina phaseolina*, and *Fusarium oxysporum*. AgNPs showed significant antioxidant efficacy with IC_50_ = 426.25 µg/mL in comparison to leaf extract with IC_50_ = 432.50 µg/mL against 2,2-diphenyl-1-picrylhydrazyl (DPPH). The AgNPs (64.36 mg AAE/g) demonstrated greater total antioxidant capacity using the phosphomolybdneum assay compared to the aqueous leaf extract (55.61 mg AAE/g) at a concentration of 1100 μg/mL. Based on these findings, AgNPs may indeed be useful for biomedical applications and drug delivery systems in the future.

## 1. Introduction

Nanotechnology is a fledgling branch of science that deals with developing as well as using nanoscale objects with distinctive physical and chemical characteristics. The typical size range of nanoparticles in each spatial dimension is between 1 and 1000 nm [1,2]. Due to changes in characteristics including shape, size, size distribution, and a larger surface area to volume ratio, nanoparticles exhibit novel and enhanced properties when compared to their bulk counterparts [3,4].

Among metal nanoparticles, silver nanoparticles (AgNPs) have gained popularity in both industry and research because of their anticancer, antimicrobial, antibiofilm, antioxidant, and anti-inflammatory effects. Nanoparticles are synthesized using physical, chemical, and biological methods. The physical methods are expensive and inefficient for the production of large-scale nanoparticles, while chemical methods involve the use of toxic chemicals. As a result, interest is growing in the safe, economical, and environment-friendly biological production of nanoparticles that uses biological reducing agents, i.e., plants, fungi, bacteria, and algae [5]. Among biological sources, the plant-based synthesis of nanoparticles is highly recommended to avoid the time-consuming task of controlling cell cultures. Moreover, various research have demonstrated that plant-mediated AgNP synthesis is safe, less toxic to organisms, easily modified, and does not require maintaining cell culture. The presence of phytochemicals such as polyphenols, flavonoids, terpenoids, aldehydes, carboxylic acids, organic acids, quinones, etc., plays an important role in the quick reduction of silver ions in the reaction mixture to synthesize AgNPs [6,7]. These bioactive compounds are present in various plant parts, including bark, roots, stems, fruits, seeds, calluses, pericarp, leaves, and flowers. Therefore, AgNPs are synthesized using a variety of plant parts [8].

Recently, microwave heating has been investigated as a potential method for synthesizing nanoparticles. The use of the microwave heating method is particularly crucial since it increases reaction kinetics, speeds up initial heating, and subsequently boosts reaction rates, resulting in clean reaction products with a quick consumption of raw materials and higher yields. Microwave irradiation rapidly and uniformly heats the reaction medium, allowing nanoparticles to develop and grow in controlled conditions. It results in uniform nucleation and growth rates, which are crucial in determining nanoparticle quality [9]. Abboud et al. reported the microwave assisted synthesis of AgNPs using aqueous onion extract (*Allium cepa*) and analyzed the influence of microwave irradiation time and microwave irradiation power on nanoparticles [10]. Joseph and Mathew reported a novel one-pot, microwave assisted method for the synthesis of AgNPs using the rhizome extract of *Alpinia galanga.* Microwave assisted synthesis resulted in small and uniform-sized nanoparticles in much lesser reaction time. The speedy consumption of starting materials reduced the formation of agglomerates and provided AgNPs with a narrow size distribution [11].

An annual herb *Trigonella foenum-graceum* L. is commonly known as Fenugreek. It’s leaves are abundant in phytochemicals that can be used to produce biogenic nanoparticles by acting as reducing and capping agents. Quercetin, rutin, ascorbic acid, trigonelline, choline, amino acids such as isoleucine, saponins, 4-hydroxyisoleucine, and histidine such as major phytochemicals in leaves have been identified by prior researchers. The ascorbic acid in leaf extract acts as a strong reducing agent. The π electrons of the double bond, the lone pair of the hydroxyl group, and the carbonyl group of the lactone ring in the ascorbic acid molecule provide sufficient reducing ability to convert Ag^+^ to AgNP. The charge transfer from ascorbic acid to Ag^+^ ions resulted in the formation of Ag atoms and, subsequently, nucleate AgNP followed by condensation and surface reduction [12,13,14]. These phytochemicals, especially polyphenols, terpenoids, and flavonoids, are also responsible for significant antioxidant and antimicrobial activities [15].

In this study, we prepared *Trigonella foenum-graceum* L. aqueous leaf extract belonging to the variety HM (Hisar Mukta) 425 using microwave heating and quantitatively analyzed its phytochemicals. Then, we reported the reduction of silver ions using leaf extract under microwave irradiation for facile and fast phytosynthesis of silver nanoparticles (AgNPs). These green AgNPs were characterized using UV-Vis spectroscopy, a particle size analyzer (PSA), FESEM-EDX (field emission scanning electron microscopy coupled to energy dispersive x-ray spectroscopy), HRTEM (high-resolution transmission electron microscopy), and FTIR (Fourier transform infrared spectroscopy). Further antioxidant activity of both the extract and AgNPs was evaluated against 2,2-diphenyl-1-picrylhydrazyl (DPPH). Additionally, the well diffusion method was used to test the antimicrobial activity against the microbial strains of *Staphylococcus aureus, Xanthomonas* spp., *Macrophomina phaseolina*, and *Fusarium oxysporum*.

## 2. Results and Discussions

### 2.1. Phytochemical Analysis of Trigonella foenum-graecum L. Aqueous Leaf Extract

Numerous phytochemicals were quantitative analyzed in *Trigonella foenum-graecum* L. aqueous leaf extract and are reported in Table 1. The total phenolic content (8.26 ± 0.13 mg Gallic acid equivalent (GAE)/g), total flavonoids (4.26 ± 0.18 mg Catechin equivalent (CE)/g), total sugars (54.48 ± 0.81 mg/g), reducing sugars (0.93 ± 0.01 mg/g), and non-reducing sugars (53.55 ± 0.80 mg/g) were reported in the aqueous leaf extract.

### 2.2. Biosynthesis of AgNPs

In general, phytochemicals in plant extract can reduce Ag^+^ ions and stabilize the as-formed AgNPs by preventing nanoparticle agglomeration via binding to metals. Using *Trigonella foenum-graceum* L. leaf aqueous extract as a reducing and stabilizing agent, we reported the green synthesis of AgNPs in this study without the use of any external reducing agents or surfactants. The methodology used was completely hazard-free, clean, non-toxic, and ecologically sustainable [16]. Quercetin, which belongs to a group of plant pigments called flavonoids, was the active constituent of *Trigonella foenum-graceum* L. leaf and might be responsible for the AgNPs synthesis [17]. According to some researchers, the -OH groups present in flavonoids such as quercetin might be responsible for the reduction of silver ions to AgNPs [18]. It is possible that the tautomeric transformation of flavonoids from enol form to keto form might release reactive hydrogen atoms that reduced silver ions to AgNPs.

The proposed mechanism of AgNP synthesis by a flavonoid reduction of silver ions to AgNPs is shown in Figure 1. The redox reaction shown below illustrates that one molecule of quercetin reduced two silver ions, as AgNO_3_ in aqueous medium dissociated into silver ions (Ag^+^) and nitrate ions (NO_3_^−^). The bond dissociation energies of the -OH bond of hydroxyl groups on the catechol moiety of flavonoids are less than other hydroxyl groups present in flavonoids [19]. Quercetin reacted with Ag^+^ as an acid and reduced it into AgNPs through more reactive hydroxyl groups attached to the carbon atoms in the aromatic ring and also provided stability against agglomeration. Aqueous leaf extract contains biomolecules that bind to metal surfaces and aid in the stabilization of nanoparticles [20].

### 2.3. Characterization of AgNPs

Using UV-Visible spectroscopy, it is possible to study the formation and stabilization of AgNPs. The surface resonance plasmon (SPR) band in biosynthesized AgNPs was located around 448 nm shown in Figure 2. The size, shape, morphology, medium’s dielectric constant, and the chemical environment of synthesized nanoparticles affects the absorption spectra of AgNPs [21]. The obtained results were in agreement with previous studies, which indicates that the *Salvadora persica* plant extract reduced silver ions into silver nanoparticles. UV-visible absorption spectra also revealed that the SPR band for AgNPs was in the 350–550 nm range [22].

The average size, size distribution profile, and polydispersity index of nanoparticles in the colloidal suspension were determined by the particle size analyzer (PSA). AgNPs had an average particle size, polydispersity index (PDI), and zeta potential of 61.23 nm, 0.21, and −30.1 mV, respectively (Figure 3). The zeta potential, which measures the surface charge of AgNPs, controls the stability of nanoparticles in an aqueous colloidal solution. The size measured using PSA was greater compared to microscopic techniques, i.e., FESEM and HR-TEM, because it measures the hydrodynamic diameter of AgNPs, which includes the phytochemical layer coated on the surface of AgNPs.

By using FESEM-EDX (field emission scanning electron microscopy coupled to energy dispersive x-ray spectroscopy) analysis, the surface morphology, size, and elemental composition of the biosynthesized AgNPs were investigated (Figure 4). The size of AgNPs in the range of 10–40 nm and spherical shape was prominent. Small- and large-sized AgNPs coexisted because of a time variation in their formation during synthesis, which showed that the formation of new nanoparticles and their aggregation occurred at the same time. The Ag weight in the EDX was determined and found to be 46.94% of the total weight, whereas the weights of chlorine (Cl), silicon (Si), carbon (C) and oxygen (O) were reported to be 5.40%, 20.86%, 11.79%, and 15.01%, respectively presented in Table 2. The EDX of synthesized AgNPs revealed the presence of silver (Ag), the residual carbon (C), chlorine (Cl), and oxygen (O) peaks, which were caused by biomolecules capped on the surface of AgNPs. The source of the element silicon’s (Si) presence in the FESEM detection was the silicon carrier.

The distribution of spherical AgNPs synthesized using *Trigonella foenum-graecum* L. leaf extract was clearly seen in the HRTEM micrographs in Figure 5a–d. The synthesized nanoparticles were nearly spherical in shape and had a uniform size distribution. The size of the AgNPs was in a range of 4–22 nm, with an average size of 15 nm. The study also demonstrated nanoparticle aggregations and physical interactions that might be due to biomolecules.

By using FTIR analysis, the various functional groups in the leaf extract and on the surface of their biosynthesized AgNPs were located. These functional groups were primarily responsible for reducing Ag^+^ to Ag and stabilizing the biosynthesized AgNPs. The analysis of the FTIR spectra of the aqueous leaf extract revealed the presence of various characteristic peaks at 3325.91, 2950.23, 2838.78, 1646.76, 1409.43, 1113.70, and 1014.15 cm^−1^, and their biosynthesized AgNPs also showed peaks at 3339.35, 2926.19, 2834.63, 1642.85, 1403.15, and 1015.91 cm^−1^, respectively (Figure 6).

The presence of -OH and -NH functional groups, respectively, or the stretching vibration of primary or secondary amines may have contributed to the peaks at 3325.91 and 3339.35 cm^−1^ in the spectra of the aqueous leaf extract and their biosynthesized AgNPs. Alkanes C-H symmetric and asymmetric stretching vibrations were attributed to the strong peaks at 2950.23, 2838.78, 2926.19, and 2834.63 cm^−1^. The peaks at 1646.76 and 1642.85 cm^−1^ demonstrated the presence of amide—C=O stretching. Smaller peaks at 1409.43 and 1403.15 cm^−1^ might have resulted from the aromatic amine group C-N stretching. It is possible to attribute the two strong, powerful peaks at 1015.13 and 1018.17 cm^−1^ to the C-O-C stretching vibrations. These two peaks, 1642.85 and 1015.91 cm^−1^, were found in AgNPs, indicating that proteins and amino acids were crucial in the reduction of Ag^+^ to AgNPs and their complexation with the surface of AgNPs. The considerable peak position shift in the FT-IR comparison spectra might be attributed to the role of leaf extracts as reducing, stabilizing, and capping agents for AgNPs.

### 2.4. Antimicrobial Activity

The antibacterial activity of *Trigonella foenum-graecum* L. aqueous leaf extracts and their biosynthesized AgNPs was tested using a well diffusion method against Gram-positive (*Staphylococcus aureus*) and Gram-negative bacterial strains (*Xanthomonas* spp.) and was compared with the control. Streptomycin and distilled water were employed as the positive and negative control, respectively. The leaf extract (1000 ppm) showed the zone of inhibition (ZOI) of 27 ± 0.19 mm and AgNPs (1000 ppm) showed a ZOI of 30 ± 0.21 mm against *Staphylococcus aureus.* The standard antibiotic streptomycin (500 ppm) showed ZOI 32 ± 0.22 mm (Figure 7). Similarly, leaf extract (1000 ppm), AgNPs (1000 ppm), and streptomycin (1000 ppm) showed ZOI 28 ± 0.2 mm, 32 ± 0.23, and 34 ± 0.25 mm, respectively, against *Xanthomonas* spp. (Figure 8).

The biosynthesized AgNPs showed higher activity against *Xanthomonas* spp. (Gram-negative bacteria) than *Staphylococcus aureus* (Gram-positive bacteria), as shown in Figure 9.

The antifungal activity of *Trigonella foenum-graecum* L. aqueous leaf extract and the biosynthesized AgNPs was tested using the well diffusion method against *Macrophomina phaseolina* and *Fusarium oxysporum* and was compared with control. Nystatin and distilled water were employed as the positive and negative control, respectively. The leaf extract (1000 ppm), AgNPs (1000 ppm), and Nystatin (100 ppm) showed ZOIs of 22 ± 0.16 mm, 23 ± 0.17, and 33 ± 0.22 mm, respectively, against *Macrophomina phaseolina* (Figure 10). The leaf extract (1000 ppm), AgNPs (1000 ppm), and Nystatin (100 ppm) showed ZOIs of 20 ± 0.14 mm, 22 ± 0.15, and 31 ± 0.21 mm, respectively, against *Fusarium oxysporum* (Figure 11). Comparison of the antifungal activity of the leaf extract and the biosynthesized AgNPs shown in Figure 12.

Although AgNPs have an antibacterial effect, their impact on microbes and antibacterial mechanisms is not well understood. Positively charged Ag ions interacted with negatively charged cell membranes, disrupting the shape of the cells and causing cell leakage, which ultimately caused cell death. The phosphorus and sulphur of the extracellular and intracellular membrane proteins are also tightly bound by AgNPs, which affects cell replication, respiration, and ultimately the cell lifespan. In addition, AgNPs can interact with the thiol and amino groups of membrane proteins, forming reactive oxygen species (ROS) that prevent cells from respirating. AgNPs’ high bactericidal activity is due to their interaction with the bacterial strain’s plasma membrane and peptidoglycan cell wall. Previous literature reported that the smaller-size AgNPs showed higher antimicrobial activities due to the larger surface area [23,24].

### 2.5. Antioxidant Activity

#### 2.5.1. DPPH Free Radical Scavenging Activity

By comparing the DPPH free radical scavenging percentage of the *Trigonella foenum-graecum* L. leaf extract and the synthesized AgNPs with standard ascorbic acid, the antioxidant activity was determined. When the sample solution was added, it was noticed that the color of the DPPH solution changed from purple to yellow. This was due to the scavenging of DPPH as a result of the hydrogen atom being donated to stabilize the DPPH molecule. The DPPH free radical scavenging activity of ascorbic acid was 92.51% at 120 μg/mL followed by 88.21, 74.18, 61.42, 48.12, and 28.54% at 100, 80, 60, 40, and 20 μg/mL concentration, respectively. The DPPH free radical scavenging activity of aqueous leaf extract was 90.42% at 1100 μg/mL followed by 86.33, 70.18, 58.55, 36.72, and 18.41% at 900, 700, 500, 300, and 100 μg/mL concentration, respectively, while the highest AgNP DPPH free radical scavenging activity was 92.81% at 1100 μg/mL followed by 88.07, 72.19, 59.43, 39.56, and 19.64% at 900, 700, 500, 300, and 100 μg/mL concentration, respectively (Figure 13). As the concentration of the extract or AgNPs increases, the DPPH free radical scavenging activity increased. Ascorbic acid had an IC_50_ of 54.33 μg/mL, while AgNPs had an IC_50_ of 426.25 μg/mL, and aqueous leaf extract had an IC_50_ of 432.50 μg/mL, indicating that ascorbic acid had the highest antioxidant efficacy. AgNPs had higher antioxidant efficacy compared to the aqueous leaf extract. This might be due to the presence of phytochemicals on the surface of AgNPs, which facilitate rapid single electron and hydrogen atom transfer, thus stabilizing the DPPH molecule [25].

#### 2.5.2. Total Antioxidant Capacity Using Phosphomolybdneum Assay

The total antioxidant capacity of the aqueous leaf extract and the biosynthesized AgNPs was estimated with the help of a standard curve using ascorbic acid. The antioxidants present in the sample had the ability to reduce molybdenum (VI) to a green colored phosphomolybdate (V) complex. The AgNPs (64.36 mg AAE/g) demonstrated greater antioxidant capacity compared to the aqueous leaf extract (55.61 mg AAE/g) at a concentration of 1100 μg/mL (Figure 14). The attachment of phenolics and flavonoids on the surface of AgNPs might be the cause of its improved antioxidant activity.

## 3. Materials and Methods

### 3.1. Chemicals and Collection of Plant Material

Himedia Private Limited provided the following chemicals: sodium carbonate, gallic acid, sodium sulphate, copper sulphate, sodium hydroxide, sodium bicarbonate, sodium potassium tartrate, sodium hydrogen arsenate, sodium phosphate, ammonium molybdate, phenol, 2,2-diphenyl-1-picrylhydrazyl (DPPH), aluminum chloride, silver nitrate (AgNO_3_), sodium nitrite, Folin–Ciocalteu reagent, Conc. sulfuric acid, catechin, nutrient agar (NA), and potato dextrose broth (PDB). The strains of *Staphylococcus aureus* (MTCC 96), *Xanthomonas* spp. (MTCC 11102), *Fusarium oxysporum* (MTCC 284), and *Macrophomina phaseolina* (MTCC 10399) were obtained from the Institute of Microbial Technology (IMTECH), Chandigarh, India’s culture collecting center for microbial resources. For the evaluation of antimicrobial activity, experiments were conducted in the department of microbiology, CCS HAU Hisar.

*Trigonella foenum-graecum* L. leaves of the Hisar Mukta (HM) 425 variety were procured from the Chaudhary Charan Singh Haryana Agricultural University’s Vegetable Science Research Farm. The collected leaf samples were verified by Dr. Anita, Assistant professor, Department of Botany and Plant physiology, CCS HAU, Hisar, India, by using an online platform (Tropicos and IPNI). The voucher specimens were verified by the Medicinal, Aromatic and Potential Crops Section, Department of Genetics and Plant Breeding, CCS HAU Hisar, by voucher specimen number 20.

### 3.2. Preparation of Trigonella foenum-graecum L. aqueous Leaf Extract

The leaves of *Trigonella foenum-graecum* L. were shade dried at room temperature. The dried leaves (5 g) and 50 mL of distilled water were taken in the flask and then microwave-irradiated (Milestone, Start S Microwave, USA, 90 W) for 5 min and allowed to cool. The filtrate was collected and utilized to synthesize AgNPs.

### 3.3. Phytochemical Analysis of Trigonella foenum-graecum L. Aqueous Leaf Extract

#### 3.3.1. Total Phenolic Content

Using the Folin–Ciocalteu method and gallic acid as the reference, the total phenolic content of the aqueous leaf extract was determined [26]. To 1 mL of extract, 1 mL of Folin–Ciocalteu reagent and 2 mL of Na_2_CO_3_ (20% *w/v*) were added, and then distilled water was used to make up a volume of 10 mL. This mixture was then allowed to stand for 8 min and was finally centrifuged at 6000 rpm for 10 min. At 730 nm, using the UV-vis double beam spectrophotometer (Model UV 1900 Shimadzu), the absorbance of the supernatant solution was measured against a blank. Similarly, the blank was prepared but instead of extract, it contained respective solvent.

#### 3.3.2. Total Flavonoids

Using an aluminum chloride colorimetric assay and catechin as a standard, the total flavonoid count was determined [27]. To 1 mL of extract, 4 mL of distilled water, 0.3 mL of 5% NaNO_2_, and, after 5 min, 0.3 mL of 10% AlCl_3_ solution were added and mixed properly. Immediately, 2 mL of 1 M NaOH was added and, using distilled water, the volume was made up to 10 mL. After mixing the solution thoroughly, at 510 nm, the absorbance of the solution was measured against a blank using a UV-vis double beam spectrophotometer (Model UV 1900 Shimadzu). Similarly, the blank was prepared, but instead of extract, it contained the respective solvent.

#### 3.3.3. Total Sugars

Using the Dubois method and D-glucose as a standard, the total sugar was determined [28]. To 1 mL of leaf extract, 2.0 mL of phenol solution was added. Then, 5.0 mL of conc. H_2_SO_4_ was poured directly in the reaction mixture, followed by the cooling of the solution for 30 min. A UV-vis double beam spectrophotometer was used to measure the absorbance of the reaction mixture at 490 nm against a blank prepared in a similar way but with the respective solvent in place of the extract.

#### 3.3.4. Reducing Sugars

Using the Nelson method and D-glucose as a standard, reducing sugars were determined [29]. To 1 mL of leaf extract, 1 mL of alkaline copper reagent was added. The solution was properly mixed, covered with aluminum foil, and heated in a hot water bath for 20–25 min. Then, it was allowed to cool at room temperature. Then, 1 mL of arsenomolybdate reagent was added to the solution, and the reaction mixture was diluted with distilled water up to a 10 mL volume. A UV-vis double beam spectrophotometer was used to measure the absorbance of the reaction mixture at 520 nm against a blank prepared in a similar way but with the respective solvent in place of the extract.

#### 3.3.5. Non-Reducing Sugars

The difference between total sugars and reducing sugars was used to determine non-reducing sugars.

### 3.4. Biosynthesis of AgNPs

In a typical experiment, 25 mL of AgNO_3_ solution (1 mM) was mixed with 0.2 mL of leaf extract solution, and then the solution was microwave-irradiated (Milestone, Start S Microwave, USA, 90 W) for 2 min. AgNP synthesis was carried out in the absence of a stabilizing agent. First, the color change in the flask containing the leaf extract and AgNO_3_ solution was observed visually. After being microwaved, it changed from being colorless to a light brown. This change in color was a typical indication that the AgNP colloidal solution formed. Centrifugation was used to remove AgNPs from the reaction mixture solution for 15 min at 10,000 rpm. The AgNPs were then dried at 37 °C and used for further analysis.

### 3.5. Characterization of AgNPs

The UV-vis absorption spectrum of AgNPs was measured using a UV-vis double beam spectrophotometer (Model UV 1900, Shimadzu) in the wavelength range of 350–550 nm. The hydrodynamic size distributions, zeta potential, and polydispersity index (PDI) of nanoparticles were calculated using the PSA Microtracnanotrac wave II equipment. The surface morphology of biosynthesized AgNPs was investigated using field emission scanning electron microscopy (JSM-7610FPlus) working at an accelerating voltage of 0.1 to 30 kV. On a JEM/2100 PLUS running at 200 kV, high-resolution transmission electron microscopy (HRTEM) was successfully completed. A drop of the biosynthesized AgNPs dissolved in ethanol was applied on a copper grid with a 400 mesh and a holey carbon film covering for the HRTEM experiments. A Perkin Elmer FT-IR spectrophotometer was used to analyze the chemical composition of the leaf extract and AgNPs.

### 3.6. Antimicrobial Activity

The antibacterial activity of the leaf extract and the biosynthesized AgNPs was assessed against *Staphylococcus aureus* (Gram-positive bacteria) and *Xanthomonas* spp. (Gram-negative bacteria) using the agar well diffusion method. Before use, the nutrient agar and petri dish were autoclaved. Then, 0.1 mL of pure bacterial culture was evenly spread on nutrient agar plates using an L-rod. Then, wells created by the well borer on agar plates were filled with 20 μL of aqueous samples of leaf extract and AgNPs (1000 ppm). Streptomycin and distilled water were employed as the positive and negative control, respectively. The plates were finally incubated at 37 °C for 24 h to obtain the results.

Using the agar well diffusion method, the antifungal activity of the leaf extract and the biosynthesized AgNPs were evaluated against *Macrophomina phaseolina* and *Fusarium oxysporum.* Before use, the potato dextrose agar and petri dish were autoclaved. Then, 0.1 mL of pure fungal culture was evenly spread on nutrient agar plates using an L-rod. The wells on the agar plates were then formed by the well borer, and 20 μL of aqueous samples of leaf extract and AgNPs (1000 ppm) was added. Nystatin and distilled water were employed as the positive and negative controls, respectively. The plates were finally incubated at 37 °C for 48 h to obtain the results.

### 3.7. Antioxidant Activity

#### 3.7.1. DPPH Free Radical Scavenging Activity

The antioxidant activity of the aqueous leaf extract and the biosynthesized AgNPs was determined using a DPPH free radical scavenging assay [30]. In a typical experiment, 1 mL of different concentrations (100–1100 μg/mL) of each sample (leaf extract, AgNPs) was mixed with 2 mL of DPPH solution (0.1 mM in methanol). After the incubation of the reaction mixture for 30 min in the dark, a UV-visible spectrophotometer was used to measure the absorbance at 517 nm. Ascorbic acid in different concentrations (20–120 μg/mL) was used as a standard and assayed in a similar manner.

The following formula was used to calculate the percentage of scavenging activity:$$\%\;DPPH\;free\;radical\;scavanging\;activity=\left[\frac{Ac-As}{Ac}\right]\times 100$$
where *Ac* is the absorbance of control, and *As* is the absorbance of sample

#### 3.7.2. Total Antioxidant Capacity using Phosphomolybdneum Assay

Using a phosphomolybdenum assay and ascorbic acid as the standard, the total antioxidant capacity of the aqueous leaf extract and the biosynthesized AgNPs was determined and expressed in milligrams of ascorbic acid equivalents per gram (mg AAE/g) [31]. In a typical experiment, 0.3 mL of different concentrations (100–1100 μg/mL) of each sample (leaf extract/AgNPs) was taken in glass vials, and3 mL of phosphomolybdenum reagent was added. Then, the solution was thoroughly mixed before being covered with lids. After incubation of the reaction mixture at 95 °C for 90 min, a UV-visible spectrophotometer was used to measure the absorbance at 695 nm. Similarly, the blank was prepared but instead of sample, it contained the respective solvent.

### 3.8. Statistical Analysis

The sample was taken in triplicate for statistical analysis. The data of total phenolics, total flavonoids, and sugars were expressed as mean standard error (±SE) using SPSS (Statistical Package for Social Sciences) version 23. Utilizing the original software, the regression analysis of the IC_50_ values for antioxidant activity was assessed.

## 4. Conclusions

Herein, we reported the green synthesis of AgNPs using *Trigonella foenum-graecum* L. leaf extract. The phytochemicals that act as reducing and capping agents in the synthesis of AgNPs were quantitatively determined. Biosynthesized AgNPs were successfully characterized using spectroscopic techniques. AgNPs showed significant antimicrobial and antioxidant activities compared to the leaf extract. Their outstanding biological activities may be due to the unique properties of nanoparticles and the adsorbed phytochemicals of the leaf extract on their surface. Additionally, this research provides a new pathway for the production of plant-derived, biocompatible nanoparticles with additional biological applications.

## Figures and Tables

**Figure 1 ijms-24-03480-f001:**
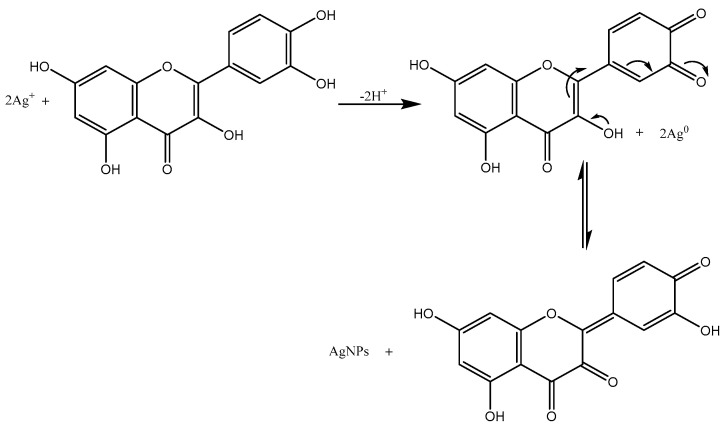
Mechanism of Ag^+^ ion reduction to silver nanoparticles (AgNPs) by quercetin.

**Figure 2 ijms-24-03480-f002:**
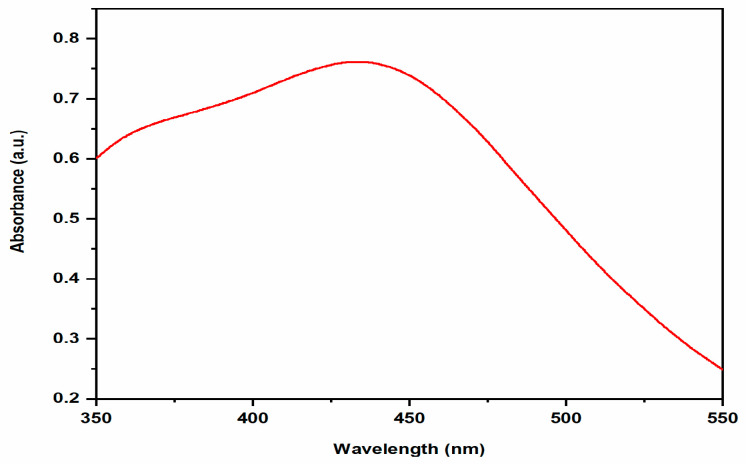
UV-vis absorption spectra of AgNPs.

**Figure 3 ijms-24-03480-f003:**
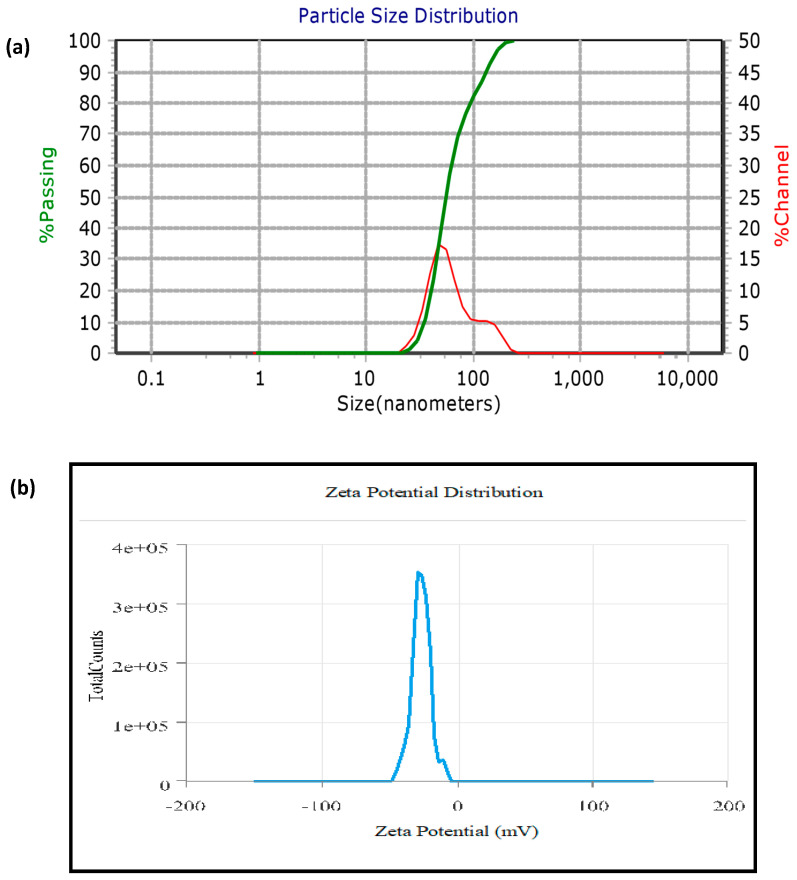
(**a**) PSA of biosynthesized AgNPs; (**b**) Zeta potential of AgNPs.

**Figure 4 ijms-24-03480-f004:**
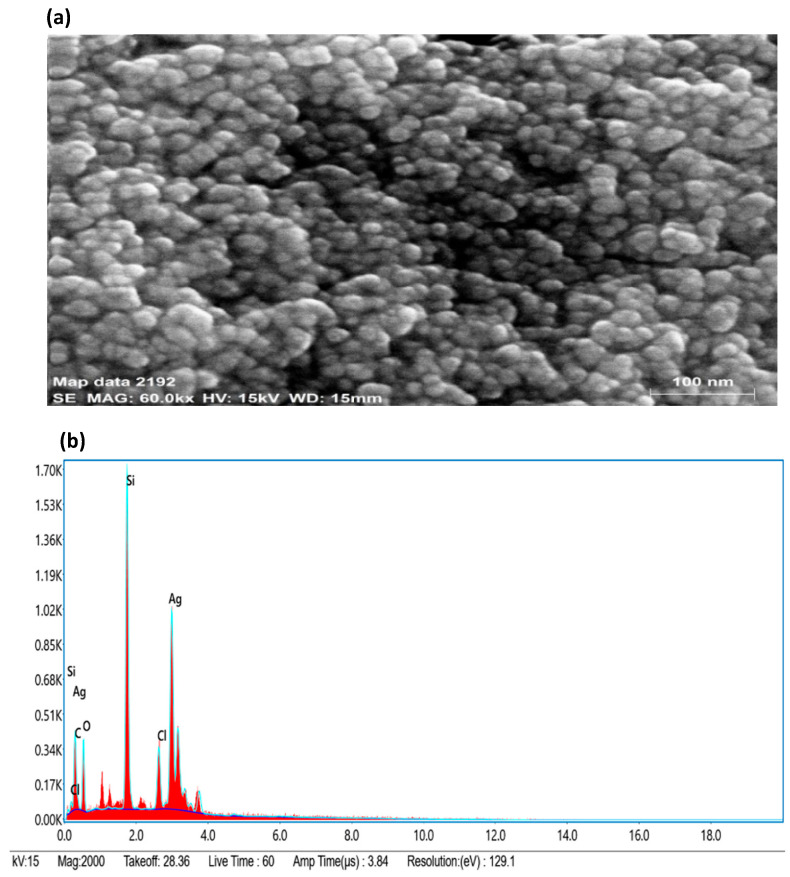
(**a**) FESEM micrograph of AgNPs at 100 nm scale; (**b**) elemental mapping of AgNPs.

**Figure 5 ijms-24-03480-f005:**
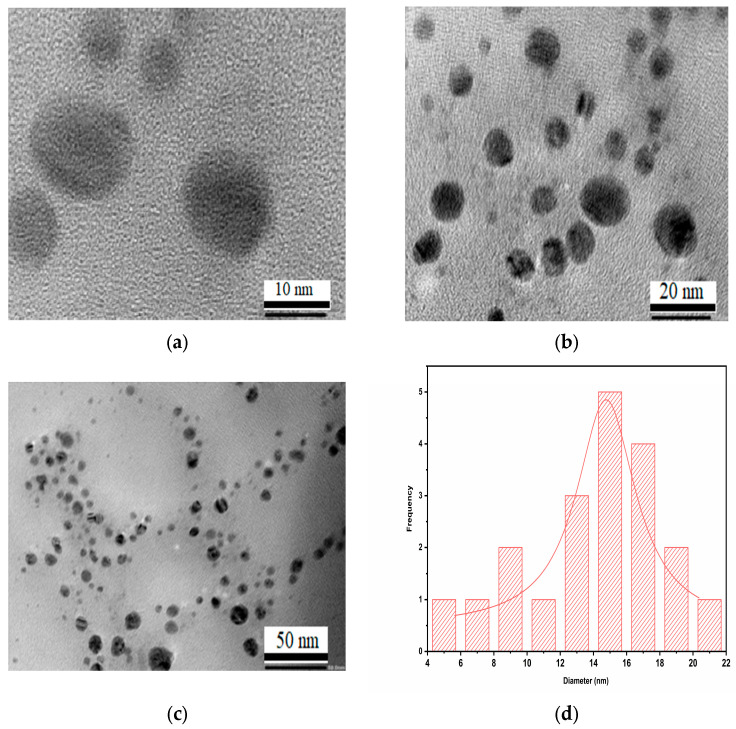
HR-TEM micrographs showing the presence of AgNPs recorded at (**a**) 10 nm, (**b**) 20 nm, and (**c**) 50 nm magnification levels; (**d**) histogram showing distributions of the size of AgNPs.

**Figure 6 ijms-24-03480-f006:**
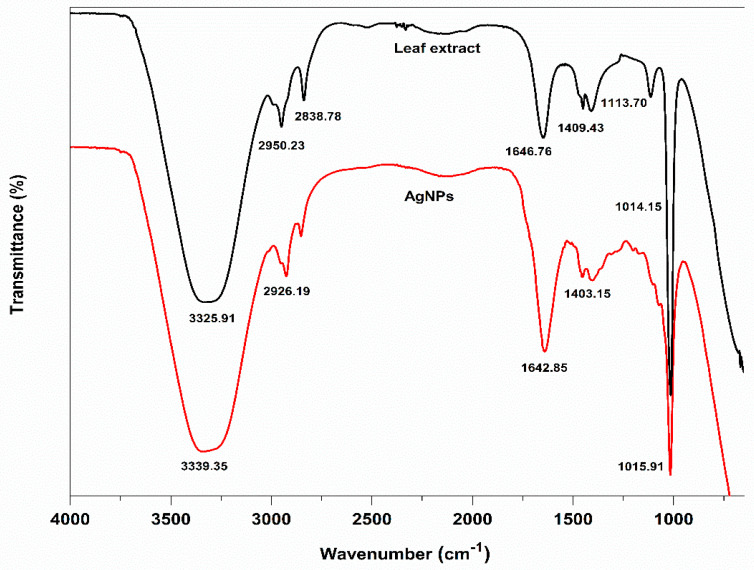
Comparative FTIR spectra of Fenugreek leaf extract (FLE) and biosynthesized silver nanoparticles (FL-AgNPs).

**Figure 7 ijms-24-03480-f007:**
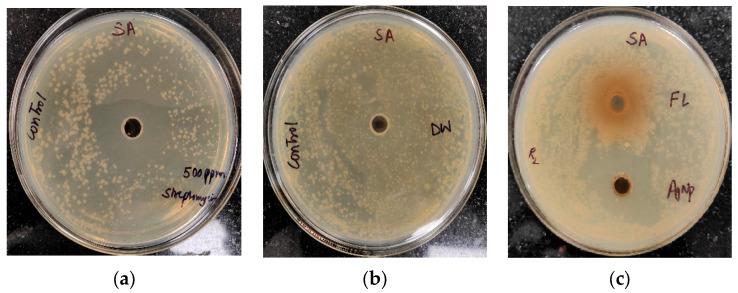
Antibacterial activity against *Staphylococcus aureus* (SA): (**a**) positive control (streptomycin); (**b**) negative control (distilled water (DW)); (**c**) leaf extract (FL) and AgNPs.

**Figure 8 ijms-24-03480-f008:**
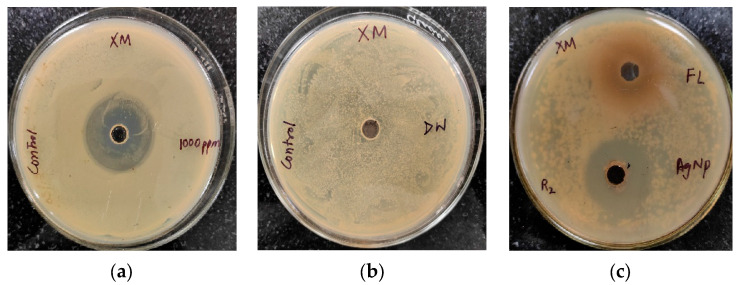
Antibacterial activity against *Xanthomonas* spp. (XM): (**a**) positive control (streptomycin); (**b**) negative control (distilled water (DW)); (**c**) Antibacterial activity against *Xanthomonas* spp. (XM) using leaf extract (FL) and AgNPs.

**Figure 9 ijms-24-03480-f009:**
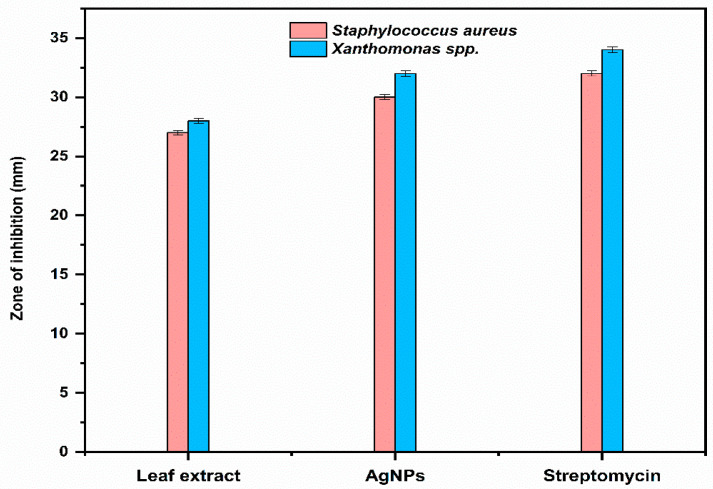
Comparison of antibacterial activity of leaf extract and the biosynthesized AgNPs.

**Figure 10 ijms-24-03480-f010:**
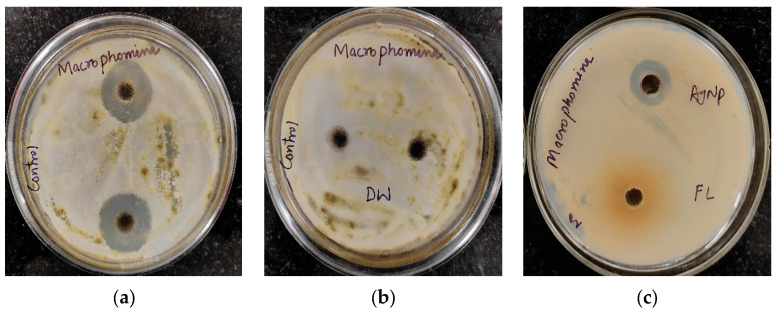
Antifungal activity against *Macrophomina phaseolina*: (**a**) positive control (Nystatin); (**b**) negative control (distilled water (DW)); (**c**) AgNPs and leaf extract (FL).

**Figure 11 ijms-24-03480-f011:**
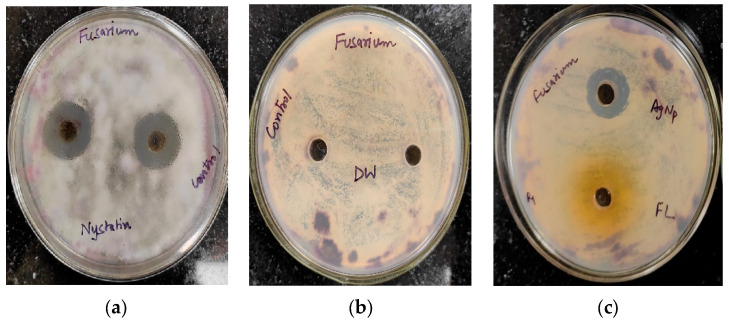
Antifungal activity against *Fusarium oxysporum*: (**a**) positive control (Nystatin) (**b**) negative control (distilled water (DW)); (**c**) AgNPs and leaf extract (FL).

**Figure 12 ijms-24-03480-f012:**
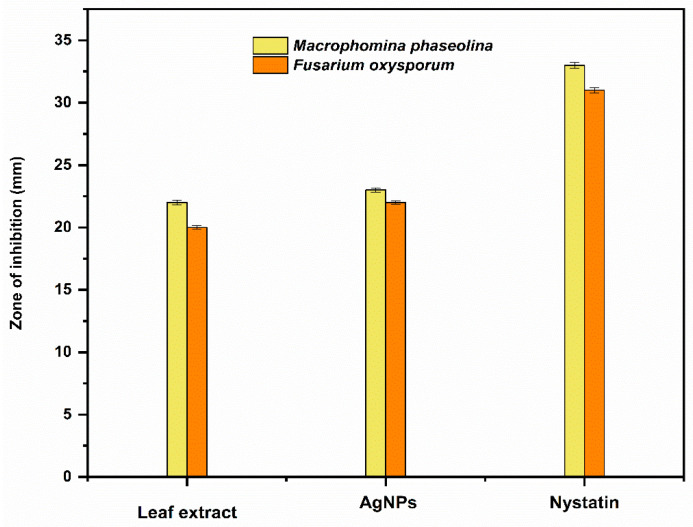
Comparison of the antifungal activity of the leaf extract and the biosynthesized AgNPs.

**Figure 13 ijms-24-03480-f013:**
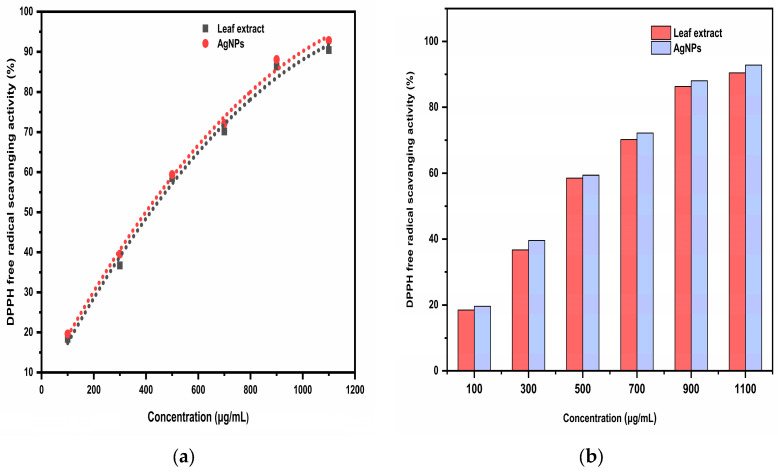
(**a**) Quadratic regression equation for the IC_50_ (µg/mL) value by DPPH free radical scavenging activity; (**b**) The comparative IC_50_ (μg/mL) of the aqueous leaf extract and the biosynthesized AgNPs.

**Figure 14 ijms-24-03480-f014:**
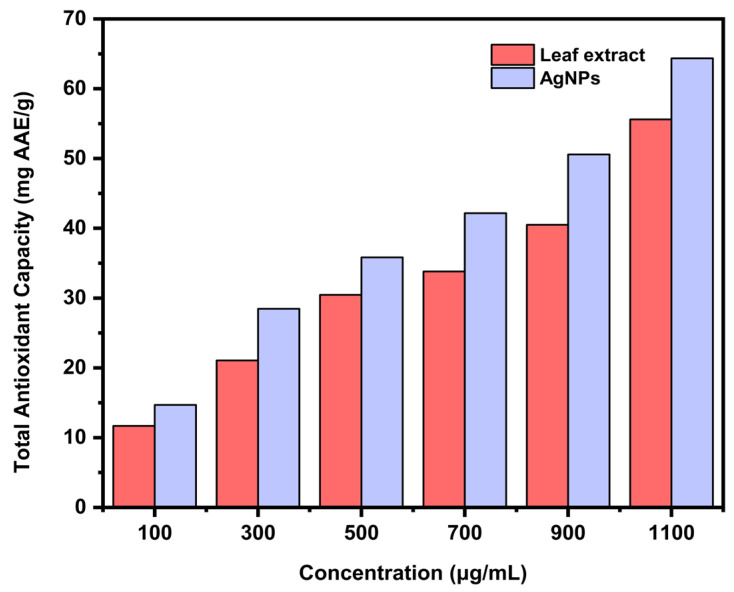
Comparative total antioxidant capacity of aqueous leaf extract and the biosynthesized AgNPs.

**Table 1 ijms-24-03480-t001:** Phytochemical analysis of *Trigonella foenum-graecum* L. (HM 425) aqueous leaf extract.

Sample	Total Phenolics (mg GAE/g)	Total Flavonoids (mg CE/g)	Total Sugars (mg/g)	Reducing Sugars (mg/g)	Non-Reducing Sugars (mg/g)
Leaf extract	8.26 ± 0.13	4.26 ± 0.18	54.48 ± 0.81	0.93 ± 0.01	53.55 ± 0.80

**Table 2 ijms-24-03480-t002:** Energy dispersive x-ray spectroscopy (EDX) elemental analysis of AgNPs.

Element	Carbon	Oxygen	Silver	Silicon	Chlorine
Weight %	11.79	15.01	46.94	20.86	5.40
Atomic %	30.21	28.87	13.39	22.85	4.68

## Data Availability

Not applicable.

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
