# Peer review of "Phytofabrication of Silver Nanoparticles Using Trigonella foenum-graceum L. Leaf and Evaluation of Its Antimicrobial and Antioxidant Activities"

_ijms, 2023, doi:10.3390/ijms24043480_

Round 1
Reviewer 1 Report
Authors presented a work related with the synthesis of silver NPs from leaf extracts. Despite it has intersting results, the quality of the manuscript is not enough for publication in IJMS.
It is neccesary to define clearly which is the novelty of the work and a deep discussion with many results, previously published in the literature, in which silver NPs are used for antimicrobial effect. Which drawbacks in the state of the art can be solved with this work? If authors do not state that, it is not possible its publication.
Main issues:
Figure 1 has disappeared. Please revise and insert it.
Figure 3... you have 2 peaks. Which integration method have you used? It is a problema of data management. Have you tried other integration methods? The value of PDI is considerably high.
Please provide Zpotential distribution as figure.
Figure 4b, more quality, it is difficult to see
Figure 5, scale is not observed
Figure 13a: x-axis is WL (nm)? please revise it
Please indicate where all reagents have been purchased
You indicate ANOVA analysis in Materials section. Where are the results? It is important to analyze if the differences in IC50 are significant or not...
Minor comments:
Abstract: please provide units for value IC50
Line 101: as-formed? what that means?
Line 117 and 134: space
Italics for strain names
Line 470: spelling diffusion
Author Response
Respected Sir,
I hereby submit point wise reply of comments for your kind consideration, please.
Authors presented a work related with the synthesis of silver NPs from leaf extracts. Despite it has interesting results, the quality of the manuscript is not enough for publication in IJMS.
It is necessary to define clearly which is the novelty of the work and a deep discussion with many results, previously published in the literature, in which silver NPs are used for antimicrobial effect. Which drawbacks in the state of the art can be solved with this work? If authors do not state that, it is not possible its publication.
Main issues:
Comment 1: Figure 1 has disappeared. Please revise and insert it.
Reply: Thanks for the comment. Done as per suggestion.
Comment 2: Figure 3... you have 2 peaks. Which integration method have you used? It is a problem of data management. Have you tried other integration methods? The value of PDI is considerably high.
Reply: Thanks for the comment. The figure for PSA of biosynthesized AgNPs is software generated figure. We have given our sample to Centre for Bio-nanotechnology (CBNT), CCS HAU HISAR for PSA analysis. This image for PSA and data are provided by CBNT.
Comment 3: Please provide Zpotential distribution as figure.
Reply: Thanks for the comment. We have given our sample to Centre for Bio-nanotechnology (CBNT), CCS HAU HISAR for PSA analysis. CBNT has not provided any figure for zeta potential distribution. It has provided only data of zeta potential, therefore we have not mentioned zeta potential distribution figure in our manuscript.
Comment 4: Figure 4b, more quality, it is difficult to see
Reply: Thanks for the comment. Done as per suggestion.
Comment 5: Figure 5, scale is not observed
Reply: Thanks for the comment. Done as per suggestion.
Comment 6: Figure 13a: x-axis is WL (nm)? please revise it
Reply: Thanks for the comment. Done as per suggestion.
Comment 7: Please indicate where all reagents have been purchased
Reply: Thanks for the comment. Done as per suggestion.
Comment 8: You indicate ANOVA analysis in Materials section. Where are the results? It is important to analyze if the differences in IC50 are significant or not...
Reply: Thanks for the comment. Yes, we have indicated ANOVA analysis in materials section. The results are reported in Table 1. For phytochemical analysis of Trigonella foenum-graecum L. (HM 425) aqueous leaf extract, we have taken sample in triplicate and the results were shown as mean standard error (S.E.). One-way and two-way analysis of variance (ANOVA) were performed (OPSTAT) only for phytochemical analysis of leaves extract.
For evaluation of antioxidant activity, we have used origin software and IC50 values are calculated from regression equation. For antioxidant activity, we have taken single sample so calculated our results using origin software.
Minor comments:
Comment 1: Abstract: please provide units for value IC50
Reply: Thanks for the comment. Done as per suggestion.
Comment 2: Line 101: as-formed? what that means?
Reply: Thanks for the comment. As-formed here means synthesized AgNPs were stabilized by phytochemicals of leaf extract.
Comment 3: Line 117 and 134: space
Reply: Thanks for the comment. Done as per suggestion.
Comment 4: Italics for strain names
Reply: Thanks for the comment. Done as per suggestion.
Comment 5: Line 470: spelling diffusion
Reply: Thanks for the comment. Done as per suggestion.

Reviewer 2 Report
1. There is another manuscript already published on the antimicrobial activity of silver nanoparticles prepared using Trigonella foenum-graecum L. leaves (Green synthesis, characterization, and antimicrobial activity of silver nanoparticles prepared using Trigonella foenum-graecum L. leaves grown in Saudi Arabia, https://doi.org/10.1515/gps-2021-0043). The author should clearly mention how there manuscript is different from the previously published manuscript and what is the novelty of their research.
2. The scientific name of the plant speecis should be itallic throughout the manuscript.
3. From where the author obtained the plant species, who identified them and its herbarium details should be mentioned in the materials and methods section.
4. Detail procedure of all experiments should be provided, citing the references only is not sufficient, especially for the phytochemical analysis, characterization, antimicrobial assays etc.
5. why the author used only 2 pathogens for the antibacterial and only 1 pathogen for antifungal assay, explain.
6. MIC and MBC should be conducted.
7. What about positive and negative control for the antimicrobial assays, the author should provide details?
8. Only one antioxidant assay is not sufficient for predicting the antioxidant potential of the AgNPs, the author should perform a few more assays.
9. All the abbreviations used in the manuscript should be explained in full form in its first occurrence.
10. positive control/reference standard should be used in the antioxidant assay.
11. Besides, a few more comments are given in the attached manuscript file, the author needs to address them.

Author Response
Respected Sir,
I hereby submit point wise reply of comments for your kind consideration, please.
Comment 1: There is another manuscript already published on the antimicrobial activity of Silver nanoparticles prepared using Trigonella foenum-graecum L. leaves (Green synthesis, characterization, and antimicrobial activity of silver nanoparticles prepared using Trigonella foenum-graecum L. leaves grown in Saudi Arabia, https://doi.org/10.1515/gps-2021-0043). The author should clearly mention how there manuscript is different from the previously published manuscript and what is the novelty of their research.
Reply: Thanks for the comment. Our manuscript is different from the previously published manuscript (https://doi.org/10.1515/gps-2021-0043) in various aspects. First, we have conducted our experiments on Trigonella foenum-graceum L. leaf belonging to specific variety Hisar Mukta (HM) 425. Second, we have first quantitative analyzed phytochemicals (total phenolic content, flavonoids, sugars) which act as reducing, stabilizing and capping agents in the reduction of silver ions to AgNPs. Third, in the previously reported manuscript, authors have prepared extract and AgNPs by simple heating but in our research we have used green technology (Microwave assisted synthesis) for preparation of leaf extract and synthesizing AgNPs. Fourth, we have also evaluated antioxidant activity of leaf extract and AgNPs.
Comment 2: The scientific name of the plant species should be italic throughout the manuscript.
Reply: Thanks for the comment. Done as per suggestion.
Comment 3: From where the author obtained the plant species, who identified them and its herbarium details should be mentioned in the materials and methods section.
Reply: Thanks for your suggestion. Trigonella foenum-graecum L. leaves of Hisar Mukta (HM) 425 variety were procured from the Chaudhary Charan Singh Haryana Agricultural University's Vegetable Science Research Farm. The collected leaves sample was verified by Dr. Anita, Assistant professor, Department of Botany & Plant physiology, CCS HAU, Hisar, India by using online platform (Tropicos & IPNI). The voucher specimens were verified by Medicinal, Aromatic and Potential Crops Section, Department of Genetics and Plant Breeding, CCS HAU Hisar by voucher specimen number 20.
Comment 4: Detail procedure of all experiments should be provided, citing the references only is not sufficient, especially for the phytochemical analysis, characterization, antimicrobial assays etc.
Reply: Thanks for your suggestion. Detailed procedure of all experiments are included in manuscript.
Comment 5: why the author used only 2 pathogens for the antibacterial and only 1 pathogen for antifungal assay, explain.
Reply: Thanks for the comment. For evaluation of antibacterial activity, we have used gram positive bacteria (Staphylococcus aureus) and gram negative bacteria (Xanthomonas spp.). For antifungal activity, Macrophomina phaseolina and Fusarium oxysporum strains were used. On basis of availability, we have conducted experiments using these strains.
Comment 6: MIC and MBC should be conducted.
Reply: Thanks for your suggestion. Yes, we agree measuring antimicrobial activity by diameter of zone of inhibition is considered a primary screening but antimicrobial activity of our sample is not so remarkable. So, we did not proceed further to find probable dose determination against specific microorganisms.
Comment 7: What about positive and negative control for the antimicrobial assays, the author should provide details?
Reply: Thanks for the comment. For evaluation of antibacterial activity, Streptomycin and distilled water are used as positive control and negative control respectively. For evaluation of antifungal activity, Nystatin and distilled water are used as positive control and negative control respectively.
Comment 8: Only one antioxidant assay is not sufficient for predicting the antioxidant potential of the AgNPs, the author should perform a few more assays.
Reply: Thanks for your suggestion. Yes, I agree that one antioxidant assay is not sufficient for predicting antioxidant potential of the AgNPs but to avoid lengthening of article, I have included one antioxidant assay by DPPH free radical scavenging activity but in my thesis I have also evaluated antioxidant assay by using phosphomolybdneum method. On your suggestion, I am including results of antioxidant assay by phosphomolybdneum method.
Comment 9: All the abbreviations used in the manuscript should be explained in full form in its first occurrence.
Reply: Thanks for the comment. Done as per suggestion.
Comment 10: positive control/reference standard should be used in the antioxidant assay.
Reply: Thanks for your suggestion. We have used Ascorbic acid as reference standard in antioxidant assay. For evaluation of antioxidant activity using DPPH free radical scavenging method, Ascorbic acid in different concentrations (20–120 μg/mL) was used as standard and it had an IC50 value of 54.33 μg/mL. Ascorbic acid is very strong antioxidant and had lower IC50 values as compared to leaf extract and their biosynthesized AgNPs. For leaf extract and AgNPs concentration range of 100-1100μg/mL were taken. Due to difference in concentration range of standard (Ascorbic acid) and sample (leaf extract/ AgNPs), we can not plot comparative graph of all three. Therefore, plotted comparative plot of leaf extract and their biosynthesized AgNPs (fig 13b).
Comment 11: Besides, a few more comments are given in the attached manuscript file, the author needs to address them.
Reply: Thanks for the comment. Done as per suggestion.

Round 2
Reviewer 1 Report
Some efforts have been done by authors in order to improve the manuscript. However, my main concern is still unclear. Where is the novelty of the work? Why this paper solve some of the drawbacks from other publications in the literature?
Authors have to clarify this before publication.
Moreover, I understand that PSD and Zpotential analysis has been carried out by external people. However, this information is essential. Try to contact them in order to include in the paper the answer of my previous comments
Author Response
Respected Editor,
I hereby submit point wise reply of reviewer’s comments for your kind consideration, please.
Comment 1:
Some efforts have been done by authors in order to improve the manuscript. However, my main concern is still unclear. Where is the novelty of the work? Why this paper solves some of the drawbacks from other publications in the literature?
Authors have to clarify this before publication.
Reply: Thanks for your suggestion.
In the present study, we have focused on green chemistry approach by including green technique i.e. microwave assisted extraction to prepare aqueous leaf extract of Trigonella foenum-graceum L. belonging to variety HM (Hisar Mukta) 425. Microwave assisted extraction has advantages over conventional methods as it shortens extraction time, less solvent consumption, high extraction yield, and also enhances the quality of extracts. Then, we have reported reduction of silver ions using aqueous leaf extract of Trigonella foenum-graceum L. belonging to variety HM (Hisar Mukta) 425 under microwave irradiation for facile and fast phytosynthesis of Silver nanoparticles (AgNPs). Synthesizing nanoparticles using microwave irradiation have become more popular by virtue of their higher heating efficiency and uniformity as compared to conventional methods of heating such as hot-plate, heating elements, or heat conduction. The use of the microwave heating method is particularly crucial since it increases reaction kinetics, speeds up initial heating, and subsequently boosts reaction rates, resulting in clean reaction products with higher yields and lower energy consumption. Microwave irradiation offers rapid and uniform heating of the reaction medium and thus provides uniform nucleation and growth conditions for nanoparticles. In conventional heating, the control over shape and size of nanoparticles have become major issue due to non- uniform reaction conditions and sharp thermal gradients throughout the bulk of solution. To the best of our knowledge, no reports related to microwave assisted synthesis of AgNPs using aqueous leaf extract of Trigonella foenum-graceum L. belonging to variety HM (Hisar Mukta) 425 are yet available.
Our manuscript is different from the previously published manuscripts (https://doi.org/10.1515/gps-2021-0043, https://doi.org/10.1016/j.mset.2021.10.001) in various aspects. First, we have conducted our experiments on Trigonella foenum-graceum L. leaf belonging to specific variety Hisar Mukta (HM) 425. Second, we have first quantitative analyzed phytochemicals (total phenolic content, flavonoids, sugars) which act as reducing, stabilizing and capping agents in the reduction of silver ions to AgNPs. Third, in the previously reported manuscript, authors have prepared extract and AgNPs by simple heating but in our research, we have used green technique (Microwave assisted synthesis) for preparation of leaf extract and synthesizing AgNPs. Fourth, we have also evaluated antioxidant activity of leaf extract and AgNPs.
Comment 2: Moreover, I understand that PSD and Z potential analysis has been carried out by external people. However, this information is essential. Try to contact them in order to include in the paper the answer of my previous comments.
Reply: Thanks for your valuable suggestion. We have contacted again Centre for Bio-nanotechnology (CBNT), CCS HAU HISAR for zeta potential figure and included the zeta potential distribution figure in our manuscript.
Reviewer 2 Report
The manuscript can be accepted in its current revised form.
Author Response
Respected Editor,
Thanks for your acceptance.